# Reactivation of Varicella Zoster Virus after Vaccination for SARS-CoV-2

**DOI:** 10.3390/vaccines9060572

**Published:** 2021-06-01

**Authors:** Mina Psichogiou, Michael Samarkos, Nikolaos Mikos, Angelos Hatzakis

**Affiliations:** 1First Department of Medicine, Medical School, National and Kapodistrian University of Athens, 11527 Athens, Greece; msamarkos@med.uoa.gr; 2Allergology Department, Laiko General Hospital, 11527 Athens, Greece; allergiologikotmemalaiko@gmail.com; 3Department of Hygiene, Epidemiology and Medical Statistics, Medical School, National and Kapodistrian University of Athens, 11527 Athens, Greece; ahatzak@med.uoa.gr

**Keywords:** SARS-CoV-2, mRNA vaccines, herpes zoster

## Abstract

Seven immunocompetent patients aged > 50 years old presented with herpes zoster (HZ) infection in a median of 9 days (range 7–20) after vaccination against SARS-CoV-2. The occurrence of HZ within the time window 1–21 days after vaccination defined for increased risk and the reported T cell-mediated immunity involvement suggest that COVID-19 vaccination is a probable cause of HZ. These cases support the importance of continuing assessment of vaccine safety during the ongoing massive vaccination for the COVID-19 pandemic and encourage reporting and communication of any vaccination-associated adverse event.

## 1. Introduction

As the pandemic of SARS-CoV-2 enters its second year and new variants of the virus emerge, vaccines are needed to protect individuals at high risk for complications and to potentially control disease outbreaks through the establishment of herd immunity [1,2]. The European Medicines Agency has already approved the emergency use of four vaccines; two of them contain a nucleoside-modified messenger RNA that encodes the viral spike (S) glycoprotein of SARS-CoV-2 and two are viral vector COVID-19 vaccines encoding the SARS-CoV-2 S glycoprotein [3]. mRNA vaccines, first-in-class tools in the biotechnological sector, have an overall effectiveness against symptomatic disease of approximately 95% for wild-type variants, while viral vector vaccines have an effectiveness close to 70% [2,3].

Surveillance of rare safety issues related to the vaccines is progressing as more data accumulate during post-marketing surveillance [4,5]. Adverse events following immunization (AEFI) may be a chance phenomenon or may be causally related to the vaccine. To answer this question, a temporal relationship with vaccination and AEFI is necessary and a biological mechanism explaining the association should be proposed. In addition, population-based studies should demonstrate increased occurrence of AEFI in vaccinated compared to non-vaccinated individuals. Post-marketing surveillance is destined to detect rare or unexpected patterns of AEFI unlikely to be recognized in pre-licensure clinical trials. Reporting of AEFI events in large databases like Vaccine Adverse Event Reporting System (VAERS) is limited by underreporting, variable data quality, the absence of defined diagnostic criteria, absent denominator information, and reporter bias [6].

In Greece, the vaccination program started in January 2021. Health care workers (HCW) accounted for a significant proportion of COVID-19 infections worldwide [7], and along with the elderly were prioritized for vaccination. 

Varicella zoster virus (VZV) is a human neurotropic virus that causes varicella. This is followed by the establishment of latency in sensory ganglia, notably, the dorsal root ganglia, trigeminal ganglia, and enteric ganglia. The latent phase usually lasts for several decades before reactivation occurs. VZV reactivation normally presents as herpes zoster (HZ), which is characterized by a painful, unilateral vesicular eruption in the dermatome innervated by the ganglion where reactivation occurred and probably appears when the immune system fails to contain the latent VZV replication. Effective treatment is available [8,9,10,11].

VZV reactivation in COVID-19 cases has been already reported. COVID-19-associated lymphopenia, especially CD3+CD8+ lymphocyte and functional impairment of CD4+ T cells, can render a patient more susceptible to developing HZ by reactivating VZV. It is also reported that HZ could be a sign of undiagnosed COVID-19 infection in younger age groups [12,13,14,15].

Here, we report on a series of patients with VZV reactivation after vaccination against SARS-CoV-2.

## 2. Materials and Methods

Seven patients (3 of 7 were women) with a mean age of 77 years (range, 51 to 94) had been initially diagnosed with HZ in primary care and they were referred to the Infectious Diseases Clinic of First Department of Internal Medicine, Laiko General Hospital, for further evaluation. All of them had been vaccinated with the Pfizer-BNT162b2 vaccine against SARS-CoV-2 and they developed HZ close to vaccination. Each case was reviewed for demographic information (patient age, sex), vaccine-related symptoms, HZ-related symptoms, and management; as well, a brief medical history (comorbidities/drugs) was obtained for everyone. Participants were followed up prospectively and a serum serology test 7–14 days after vaccination was taken and tested for anti-SARS CoV-2 receptor binding domain (RBD) spike protein. 

## 3. Results

Patients’ characteristics are shown in Table 1. All patients were >50 years old, without risk factors or comorbidity that would contribute to the development of HZ, such as immunosuppressive drug use, radiation therapy, trauma, or psychological stress. No patient had been vaccinated against varicella or reported a previous episode of HZ, although all of them reported varicella infection during infancy. They all received the Pfizer-BNT162b2 vaccine against SARS-CoV-2 between 16th January and 15th February of 2021. Mild local symptoms or mild fatigue occurred in 4 out of 7 patients. They developed HZ in a median of 9 days (range 7–20 days) after the administration of the most recent dose (first or second). In five patients, HZ occurred between 7 and 20 days after the first dose and in 2 patients between 7 and 9 days after the second dose. Prodromal pain was reported in 5 patients. In all patients a single dermatome was affected; however, two patients developed HZ on the second branch of the trigeminal nerve. Reverse transcriptase-polymerase chain reaction for SARS-CoV-2 from nasopharyngeal swabs was negative in all patients. In 3/7 patients the episode was classified as severe, i.e., it led to zoster-related hospitalization, the pain was severe as estimated by a McGill pain questionnaire [16], or a neuropathic pain medication was necessary for HZ-associated pain. In 6 cases, treatment with oral valacyclovir led to resolution of the lesions after 10 days. The patient with herpes ophthalmicus received intravenous acyclovir and required hospitalization. Three patients had subacute herpetic neuralgia which required medication for neuropathic pain. Guidance regarding the second dose of vaccine when the eruption occurred between the two vaccine doses has varied. The physicians who treated the 7th patient with a severe HZ clinical presentation advised the patient not to receive the 2nd dose of the vaccine. 

## 4. Discussion

We present a series of 7 cases of HZ in immunocompetent individuals that occurred in a mean (range) of 9 days (7–20 days) after COVID-19 vaccination. The major risk factor for reactivation of VZV is increasing age, partly because of age-related decline in specific cell-mediated immune responses to VZV, while other risk factors include disease-related immunocompromise such as HIV infection, iatrogenic immunocompromission, physical trauma, or comorbid conditions such as malignancy or chronic kidney or liver disease [17,18,19]. The most common complication of HZ is postherpetic neuralgia, while other complications include ocular ones (HZ opthalmicus, acute retinal necrosis), neurological ones (Ramsay Hunt syndrome, Bell’s palsy, aseptic meningitis, encephalitis, peripheral motor neuropathy, myelitis, Guillain-Barre syndrome, stroke syndromes), and bacterial infection of the skin [17]. Approximately 8–10% of patients with HZ experience complications in addition to pain, while the incidence of recurrence is 10.96 per 1000 person-years with an average of 5.6 years of follow-up [9]. One to six percent of patients will experience a second episode of HZ. Cell-mediated immunity is critical for the maintenance of latency and to limit the potential for reactivation, while recurrences are more common among patients with immunosuppression [18]. The severity of disease, and the risk of complications related to the reactivation, also increases with age [17]. Three of the reported patients had severe HZ requiring neuropathic medication for subacute herpetic neuralgia and one required hospitalization for ocular herpetic disease. 

The safety profile of BNT162b2 mRNA vaccine was characterized by short-term, mild-to-moderate pain at the injection site, fatigue, and headache, while the incidence of serious adverse events was low and similar in vaccine (0.6%) and placebo (0.5%) groups [1]. Overall, more vaccine than placebo recipients reported any adverse event (27% and 12%, respectively) or a related adverse event (21% and 5%) reflecting the reactogenicity events which were reported as adverse events more commonly by vaccine recipients. HZ was not reported as an adverse event among the vaccine recipients in this trial for a median period of 2 months after the second dose. Adverse events, other than local and systemic reactogenicities, were categorized as related, severe, or life threatening without specific mention of HZ [1]. Moreover, the study includes 2 months of follow-up in half of the participants [1].

The incidence rate of HZ increases with age from 3.1 per 1000 person-years for persons 45–54 years old, to 5.7 for persons 55–64 years old, to 11.8 per 1000 person-years for persons older than 65 years. The incidence rate of HZ among the unvaccinated-for-VZV and immunocompetent population > 50 years old is 9.92 (95%CI, 9.82–10.01) per 1000 person-years or 0.83 (95% CI 0.82, 0.84) per 1000 person months, according to the Kaiser Permanente Southern California study [9]. Vaccine trials do not include the person-time to estimate the incidence of rare events [1,20,21]. 

Concerning the mRNA -1273 SARS-CoV-2 vaccine, safety assessments included adverse events for 28 days after each injection [21]. So, for individuals in the placebo group, aged > 65 years old (*n* = 3763), the expected HZ cases is 3, based on the Kaiser Permanente Study [9]. However, no HZ cases were observed in vaccine and placebo recipients. During the observation period of the trial, a slight excess of Bell’s palsy occurred in the vaccine group (3 participants; <0.1%) compared to the placebo group (1 participant; <0.1%), arousing concerns that it may be more than a chance event—a possibility requiring close monitoring [21]. HZ is one of the common viral infections associated with facial palsy. In a large series of 1701 cases of Bell’s palsy, 116 individuals (6.8%) had HZ [22]. Moreover, a proportion of patients with Bell’s palsy have Ramsay Hunt syndrome (zoster sine herpete), a diagnosis based on either a fourfold rise in antibody to VZV or the presence of VZV DNA in auricular skin, blood mononuclear cells, middle ear fluid, or saliva [23]. 

The incidence of HZ among individuals with HIV infection increases with decreasing CD4 cell counts, highlighting the importance of T-cell immunity in maintaining latency of VZV [19,24]. 

Post-vaccination HZ is rarely reported in the literature. Walter et al. [25] reported three different cases of herpes virus reactivation following inactivated influenza, hepatitis A, and rabies and Japanese encephalitis vaccines, while Bayas et al. [26] reported a case of branchial plexus zoster after yellow fever vaccination. Rothova et al. [27] reported another complication of varicella-zoster infection, namely acute retinal necrosis. In this case the patient had a previous episode of VZV-associated acute retinal necrosis and presented a new episode after vaccination with the 2009 H1N1 influenza vaccine.

Regarding COVID-19 vaccination, we could only retrieve a publication by Bostan et al. [28], who reported a case of VZV reactivation in a 79-year-old patient 5 days following vaccination and a more recent one with six immunocompromised patients with autoimmune inflammatory diseases who developed a first episode of HZ closely after vaccination with the BNT162b2 mRNA vaccine [29]. 

Despite the rarity of publications, HZ is not a rare event. In the VAERS, when the COVID-19 and the varicella vaccine were excluded, there were 1653 reports of HZ-related complications from July 1990 to March 2021. The HZ-related adverse events reported for the COVID-19 vaccines were 232 [30]. In the Yellow Card adverse reaction reporting scheme of Medicines and Healthcare products Regulatory Agency of the United Kingdom (MHRA), as of 21 March 2021, there were 331 reports of HZ after the Pfizer/BioNTech vaccine and 297 after the Astra-Zeneca vaccine [31,32,33]. However, these reports could possibly underestimate the incidence of HZ, because HZ is being underreported as an adverse event following immunization for other infectious agents. 

Reactivation of VZV is a failure of the T cell compartment to maintain control of the infection. This is supposed to occur more frequently with increasing age due to adaptive immunosenescence [34]. On the other hand, a vaccine strongly stimulates the immune system and polarizes it to a vaccine-induced T cell response. In healthy adults, as Sahin U et al. reported [35], vaccination with BNT162b2 induces a co-ordinated humoral and cellular adaptive immunity. Seven days after the booster dose, a strong cellular response with spike-specific CD8+ T cell and T helper type 1 (Th1) CD4+ T cells is expanding with a high fraction of them producing interferon-γ (IFNγ), a cytokine responsible for several antiviral responses. The magnitude of S-specific CD4+T cell responses correlated positively with S1-binding IgG and also with the strength of S-specific CD8+ T cell responses that also correlated positively with S1-binding IgG. Moreover, the SARS-CoV-2 mRNA-1273 vaccine elicited a strong CD4 cytokine response involving type 1 helper T cells among participants older than 55 years old [36].

Vaccine-induced reactivation of HZ may have similarities with immune reconstitution inflammatory syndrome (IRIS), which is a paradoxical worsening of preexisting infection unmasked by the host’s regained capacity to mount an inflammatory response following the initiation of ART [24]. We postulate that VZV-specific CD8+ cells are not, temporarily, capable of controlling VZV after a massive shift of naïve CD8+ cells to produce CD8+ cells specific to control HIV or VZV. 

AEFI may be coincidental events that have occurred by chance or the vaccine may have increased the risk of the specific adverse event [6,37]. Standard algorithms have been developed by WHO and CDC to conduct causality assessments of individual cases of AEFI. “Consistent with” a causal relationship means that the event occurred during the time window defined for increased risk. Moreover, according to WHO modified criteria on the definitions for causality, a “probable” association means a temporal relationship and the existence of a biologic mechanism for causal association between the vaccine and the event [6,37,38]. HZ in this case series occurred during the time window defined for known increased risk (1–21 days after the priming dose) [39], so the AEFI can be classified as “consistent with”, and a “probable” causal association based on the World Health Organization Working Group may be proposed, reflecting the existence of temporal association and a plausible biologic mechanism [6,37,38,39]. 

Given that the decline in immune responses in the older population has been well documented and the severity of viral and bacterial infections is increased among older individuals, vaccination serves as the main strategy for preventing such infections or their severity by inducing efficient cellular and humoral responses. On the other hand, immunosenescence is often responsible for failing to induce long-term protective immunity after vaccination and offer limited duration of protection and so further studies are warranted to improve understanding of the immunological mechanisms that govern durable protection against SARS-CoV-2 and VZV. Even though the case fatality of HZ is extremely low [9], HZ is frequently associated with disability, especially among older individuals, and available treatment modifies only the severity and the duration of pain, decreases viral shedding reducing the risk of transmission, and prevents post-herpetic neuralgia. The recombinant zoster vaccine is recommended in immunocompetent adults aged more than 50 years old as it provides greater protection against HZ than the live attenuated vaccine; however, it is not widely available [40]. Clinicians may not be prepared to correlate HZ with vaccination for COVID-19. The awareness of this clinical condition encourages additional reporting and communication of HZ after vaccination. Moreover, these data might be clinically useful for considering transient prophylaxis with valacyclovir prior to vaccination for those patients at higher risk for reactivation of VZV following vaccination for SARS-CoV-2. 

## 5. Conclusions

Huge vaccination programs are ongoing. Post-marketing surveillance systems must be in place and continuing assessment of vaccine safety is important for the detection of any event that could attenuate the expected benefits, and thus to take any necessary action to minimize risks to vaccinated individuals. HZ seems to be a “probable’’ although rare AEFI, based on the criteria of temporal association with vaccination and a plausible biological association. In view of the hundreds of millions of individuals to be vaccinated for SARS-COV-2, a potential causal association may result in a large number of cases with potential severe complications among the elderly. Fortunately, HZ is a treatable situation and clinical review committees should decide whether VZV treatment is temporarily recommended before initiation of vaccination. In addition, mRNA platform is a promising new technology and demonstration of its safety is meaningful for the development of further vaccines against viruses and tumors [41].

## Figures and Tables

**Table 1 vaccines-09-00572-t001:** Demographics, SARS-CoV-2 vaccination timing and side effects, varicella zoster characteristics and management of reported patients.

Variable	Patient 1—HCW	Patient 2—HCW	Patient 3—HCW	Patient 4	Patient 5	Patient 6	Patient 7
Age (yrs.)	51	56	69	86	90	91	94
Sex (F/M)	F	F	F	M	M	M	M
Adverse events (AE) from vaccine (pain, oedema, fever, fatigue, other)	-1st dose [9 February 2021], no AE -2nd dose [2 March 2021] mild pain oedema for 2 days	-1st dose [1 February 2021] Mild pain, mild fatigue for 1 day	-1st dose [29 Jnauary 2021] mild pain for 1 day -2nd dose [8 March 2021], no AE	-1st dose [23 January 2021], no AE -2nd dose [13 February 2021], no AE	-1st dose [22 January 2021], no AE -2nd dose [12 February 2021], no AE	-1st dose [16 January 2021], no AE -2nd dose [26 February 2021], no AE	-1st dose [15 February 2021], mild pain, mild oedema for 2 days
Comorbidities/Drugs	No/No	Osteoporosis, Dyslipidemia/No	Mitral valve surgery/antiarrhythmic drug, acetylsalicylic acid	Prostate cancer, Hypertension/long-acting GnRH agonist, antihypertensive drugs	Hypertension, COPD, hyperuricemia/antihypertensive drugs, inhalers, hypozuric drugs	Dyslipidemia, hypertension/antihypertensive, antilipemic drugs	HF NYHA 3 *, pacemaker, CKD/drugs for HF
Varicella Zoster							
Days of VZ onset after vaccination	9 days after 1st dose	14 days after 1st dose	8 days after 1st dose	7 days after 2nd dose	9 days after 2nd dose	7 days after 1st dose	20 days after 1st dose
Prodromal pain days prior to exanthema (Yes/No)	Yes, 2 days	Yes, 5 days	Yes, 7 days	No	No	Yes, 7 days	Yes, 1 day
Dermatome	Lumbar	Thoracic	Fifth cranial nerve	Thoracic	Thoracic	Fifth cranial nerve, Herpes zoster opthalmicus	Thoracic
Other symptoms (headache, fever, malaise, fatigue)	Malaise, ischial pain	Malaise	Malaise, headache	No	No	Fatigue	Fatigue
Symptom’s severity	Moderate	Mild	Severe	Mild	Mild	Severe	Severe
Treatment for HZ, [Yes/No (days)]	Yes, valacyclovir, 2 days after symptom onset	Yes, valacyclovir, 5 days after symptom onset	Yes, valacyclovir, 7 days after symptom onset	Yes, valacyclovir same day	Yes, valacyclovir, same day	Yes, acyclovir IV and then valacyclovir, same day	Yes, valacyclovir, same day
Hospitalization, [Yes/No (days)]	No	No	No	No	No	Yes, 14 days	No
Postherpetic neuralgia (Yes/No)	No	No	Yes, neuropathic drugs	No	No	Yes, neuropathic drugs	Yes, neuropathic drugs
Pain scale (1–10)	6	3	4	4	3	4	5
Previous vaccination for VZV (Yes/No)	No	No	No	No	No	No	No
History of previous HZ (Yes/No)	No	No	No	No	No	No	No
Antibodies IgG to SARS-CoV-2 (Yes/Not Done))	Yes	Yes	Yes	Yes	ND	ND	ND

* (NYHA) New York Heart Association functional class.

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
