# Peer review of "Reactivation of Varicella Zoster Virus after Vaccination for SARS-CoV-2"

_vaccines, 2021, doi:10.3390/vaccines9060572_

Round 1

Reviewer 1 Report

Well presented and timely. I have few queries and suggestions for improvement.

Line 14 - add " after probable

Line 41-43. Does this mean that HCWs prioritised along with the elderly for vaccination. Could add 'also' in line 43 - were also priority...

Intro: which vaccine(s) were given? BNT162b2 and mRNA-1273 are mentioned, but readers (like me) would like to know whether this is Pfizer, Astra-Zeneca or what vaccine. 

Table on p.2-3. The right hand column 'Patient 5' needs deletion for the two peculiar inserts. What does 'NYHA' in Patient 7 column mean?

Do you have patient T-cell lymphocyte counts to add to the Table?: much is made later about low T-cell counts as a risk factor for HZ.

Line 138-145. Do you need this paragraph (about HIV)? as it adds nothing to the discussion.

The conclusion over relevance of vaccine to clinical HZ is carefully presented. Further time will tell.

Author Response

Reviewer: 1

Line 14 - add " after probable

Response: We added it.

Line 41-43. Does this mean that HCWs prioritised along with the elderly for vaccination. Could add 'also' in line 43 - were also priority...

Response: Yes, HCWs and the elderly were the first groups for vaccination. We changed it as proposed (lines 42-44).

Intro: which vaccine(s) were given? BNT162b2 and mRNA-1273 are mentioned, but readers (like me) would like to know whether this is Pfizer, Astra-Zeneca or what vaccine. 

Response: The presented patients received the Pfizer-BNT162b2 vaccine. The information is mentioned in line 64.

Table on p.2-3. The right-hand column 'Patient 5' needs deletion for the two peculiar inserts. What does 'NYHA' in Patient 7 column mean?

Response: The right-hand column was deleted, and Table 1 was modified accordingly. The Table is presented in a landscape orientation.

NYHA is New York Heart Association (NYHA) functional class. The information was added as a footnote in Table 1 (line 98).

Do you have patient T-cell lymphocyte counts to add to the Table?: much is made later about low T-cell counts as a risk factor for HZ.

Response: No, we did not have such data. However, all patients were immunocompetent with lymphocytes counts within normal range.

Line 138-145. Do you need this paragraph (about HIV)? as it adds nothing to the discussion.

Response: We deleted most of the information in lines 138-145 (new 154-159).

The conclusion over relevance of vaccine to clinical HZ is carefully presented. Further time will tell.

Response: Thank you for your comment.

Reviewer 2 Report

this study is quite interesting during this covid-19 pandemic situation. people are concerned about the side effect or many adverse events after vaccination.

there are a few of my suggestions/concerns 

  1. "Materials and Methods" this section does not provide any significant research done by the authors. it should contain the woks or analysis done by the authors
  2. table 1 "Patients with HZ close to vaccination for COVID-19" should be rewritten in a self-explanatory way.
  3. table 1 should be rearranged, (patient 5 is two times mentioned, little confusing)
  4. conclusion should be concise and focussed
  5.  

Author Response

Reviewer 2

This study is quite interesting during this covid-19 pandemic situation. People are concerned about the side effect or many adverse events after vaccination.

there are a few of my suggestions/concerns. 

1. "Materials and Methods" this section does not provide any significant research done by the authors. it should contain the woks or analysis done by the authors.

Response: We added the work and analysis done by the authors (lines 65-70).

2. table 1 "Patients with HZ close to vaccination for COVID-19" should be rewritten in a self-explanatory way.

Response: we rewrote the title of Table 1

3. table 1 should be rearranged, (patient 5 is two times mentioned, little confusing)

Response: We deleted it.

4. conclusion should be concise and focussed

Response: This comment contradicts with Reviewer 1. However, we deleted the first phrase.

Reviewer 3 Report

Psichogiou et al. report nine cases of elderly people with herpes zoster reactivation following COVID vaccination. The manuscript is well-written and can in principle be published. I have only minor remarks.

  1. The sentence lines 215ff should be rewritten - it does not make sense to me.
  2. There is a recent publication that the authors should cite, PMID  33937467, which reports five cases of VZ reactivation after Pfizer/Biontech COVID vaccination, which supports the findings presented here.

Author Response

Reviewer 3
Psichogiou et al. report nine cases of elderly people with herpes zoster reactivation following COVID vaccination. The manuscript is well-written and can in principle be published. I have only minor remarks.

  1. The sentence lines 215ff should be rewritten - it does not make sense to me.

Response: We rephrased the sentence.

  1. There is a recent publication that the authors should cite, PMID  33937467, which reports five cases of VZ reactivation after Pfizer/Biontech COVID vaccination, which supports the findings presented here.

Response: We tried to find this paper PMID  33937467 but was not possible. After searching PubMed we found a new paper published with 6 cases of HZ after Pfizer/BioNTech COVID vaccination in patients with autoimmune inflammatory rheumatic diseases and we added it in the text (lines 172-174) and reference 29.

Reviewer 4 Report

This is a clearly designed and well written report on clinical VZV reactivation after Covid vaccination (Pfizer RNA vaccine).

Criteria for causal association of AEFIs with previous vaccination are well summarized in the introduction and applied to Covid vaccine in the discussion section.

The report is of general interest and of specific relevance to the ongoing Covid vaccination, especially for aged people worldwide.

Authors should specify in more detail what are they suggesting for Zoster prevention before vaccination. There are two ways: 1) administration of Valacyclovir around the time of vaccine administration (2 separated doses), 2) preliminary vaccination against VZV. Both interventions carry considerable costs. This could be an opportunity to ask for an acceptable cost of Zoster vaccination in fragile/aged people. In fact, generally, VZV vaccination is not taking off, principally due to high costs.

Line 86: one patient advised not to receive it (please clarify).

Line 93: immunocompromission

Line 154 etc.: Bostan E et al. Please omit initials for references in the text.

Line 179: define IRIS

Line 228: development of further vaccines against viruses and tumors.

Minor English typos to be corrected.

Author Response

Reviewer 4
This is a clearly designed and well written report on clinical VZV reactivation after Covid vaccination (Pfizer RNA vaccine).

Criteria for causal association of AEFIs with previous vaccination are well summarized in the introduction and applied to Covid vaccine in the discussion section.

The report is of general interest and of specific relevance to the ongoing Covid vaccination, especially for aged people worldwide.

Response: Thank you for your comments.

Authors should specify in more detail what are they suggesting for Zoster prevention before vaccination. There are two ways: 1) administration of Valacyclovir around the time of vaccine administration (2 separated doses), 2) preliminary vaccination against VZV. Both interventions carry considerable costs. This could be an opportunity to ask for an acceptable cost of Zoster vaccination in fragile/aged people. In fact, generally, VZV vaccination is not taking off, principally due to high costs.

Response: We added our thesis in lines 227-232.

Line 86: one patient advised not to receive it (please clarify).

Response: We clarified it (Lines 91-93)

Line 93: immunocompromission

Response: We changed it.

Line 154 etc.: Bostan E et al. Please omit initials for references in the text.

Response: we deleted it.

Line 179: define IRIS

Response: We defined IRIS (lines196-199).

Line 228: development of further vaccines against viruses and tumors.

Response: We changed the phrase as suggested.

Minor English typos to be corrected.

Response: We corrected accordingly.

Thank you for your comments.
